# Lung Microbiome Participation in Local Immune Response Regulation in Respiratory Diseases

**DOI:** 10.3390/microorganisms8071059

**Published:** 2020-07-16

**Authors:** Juan Alberto Lira-Lucio, Ramcés Falfán-Valencia, Alejandra Ramírez-Venegas, Ivette Buendía-Roldán, Jorge Rojas-Serrano, Mayra Mejía, Gloria Pérez-Rubio

**Affiliations:** 1HLA Laboratory, Instituto Nacional de Enfermedades Respiratorias Ismael Cosío Villegas, Mexico City 14080, Mexico; li260387@uaeh.edu.mx (J.A.L.-L.); rfalfanv@iner.gob.mx (R.F.-V.); 2Tobacco Smoking and COPD Research Department, Instituto Nacional de Enfermedades Respiratorias Ismael Cosío Villegas, Mexico City 14080, Mexico; aleravas@hotmail.com; 3Translational Research Laboratory on Aging and Pulmonary Fibrosis, Instituto Nacional de Enfermedades Respiratorias Ismael Cosío Villegas, Mexico City 14080, Mexico; ivettebu@yahoo.com.mx; 4Interstitial Lung Disease and Rheumatology Unit, Instituto Nacional de Enfermedades Respiratorias Ismael Cosío Villegas, Mexico City 14080, Mexico; jorroser@gmail.com (J.R.-S.); medithmejia1965@gmail.com (M.M.)

**Keywords:** lung microbiome, 16S rRNA gene, immune response, respiratory diseases, dysbiosis

## Abstract

The lung microbiome composition has critical implications in the regulation of innate and adaptive immune responses. Next-generation sequencing techniques have revolutionized the understanding of pulmonary physiology and pathology. Currently, it is clear that the lung is not a sterile place; therefore, the investigation of the participation of the pulmonary microbiome in the presentation, severity, and prognosis of multiple pathologies, such as asthma, chronic obstructive pulmonary disease, and interstitial lung diseases, contributes to a better understanding of the pathophysiology. Dysregulation of microbiota components in the microbiome–host interaction is associated with multiple lung pathologies, severity, and prognosis, making microbiome study a useful tool for the identification of potential therapeutic strategies. This review integrates the findings regarding the activation and regulation of the innate and adaptive immune response pathways according to the microbiome, including microbial patterns that could be characteristic of certain diseases. Further studies are required to verify whether the microbial profile and its metabolites can be used as biomarkers of disease progression or poor prognosis and to identify new therapeutic targets that restore lung dysbiosis safely and effectively.

## 1. Introduction

The human microbiome is defined as the collection of microorganisms (archaea, viruses, bacteria, and fungi) and their genes that inhabit a body part [1]. In 2008, the human microbiome project (HMP) was created to analyze its participation in health and disease. Initially, the study was performed in the nasal and oral cavities, skin, genitourinary tract, and gut [2]. The first results of the HMP do not include the lung because of the old dogma that the “lungs were sterile tissues;” nevertheless, since 2003, there has been evidence that the respiratory tract of patients with cystic fibrosis (CF) hosted bacteria [3]. Afterwards, in 2010, the first study of 16S rRNA microbiome genes comparing healthy individuals, patients with asthma, and those with chronic obstructive pulmonary diseases (COPD), was published, reporting that there is a distinctive microbiome in healthy individuals that differs from patients with lung disease [4,5]. Currently, we know that the lung is an organ that is exposed continuously to microorganisms. In healthy subjects, the similarity of the lung and mouth microbiota varies considerably; it is undetermined whether this variation is persistent over time within subjects or is correlated with clinical phenomena such as esophageal reflux, laryngeal dysfunction, and oral hygiene [6,7]. Metagenomics techniques have demonstrated that in healthy subjects, microorganisms include viruses (virome) [8], fungi (mycome), and bacteria (bacteriome) have an important role in homeostasis [9]. These also participate in host protection against pathogenic microorganisms [10]. Interactions between commensal bacteria and immunological barriers, such as genitourinary and intestinal mucosa, have been previously found in different tissues [11]. It is not unexpected that the respiratory luminal ecosystem shares features with these ecologic niches regardless of their particular characteristics, such as an enclosed mucous tissue system structure, a high content of phospholipids, products derived from gas exchange, specific immunoglobulins [12], oxygen tension, differences in blood flow, pH, and temperature [6], ciliary clearance and other mechanic phenomena [13].

Scientific literature has marked the importance of the microbiome in disease because there are promising results in the understanding of the pathologic process and pathobiome; however, the number of lung microbiome reports is low in comparison to other ecological niches such as the gut (Figure 1). In 2010, almost 500 papers on the gut microbiome and associated diseases were available in “PubMed,” and in 2019, nearly 4000 publications were accessible, while at the same time, studies on the lung microbiome and diseases that affect this organ have accumulated to nearly 150 publications.

Considering the importance of the microbiome in immune regulation and its participation in diseases, we aimed to describe the immunological mechanism by which the lung microbiome participates in respiratory diseases, as well as the clinical implications that have been proposed.

## 2. Materials and Methods

A literature review was performed using the “PubMed” database; the following Medical Subject Heading terms were used: “Microbiota” OR “Metagenome” AND “Lung Disease” OR “Lung” AND “Immune System” OR “Immunomodulation” OR “Immunity.” Studies in all languages published between January 2015 and December 2019 were included. Selection criteria included original articles in which 16S rRNA (bacteriome) sequencing for human and animal models was employed. For the bibliometric analysis, the package used was RISmed [14], and for the creation of a proximity matrix using multidimensional scaling (MDS), we used Bibliometrix [15], in RStudio V 1.2.1335 IOS [16], following the workflow proposed by the library’s developer. We used ggplot2 for the creation of graphics [17].

## 3. Results

The article selection process was carried out following the PRISMA guidelines (Preferred Reporting Items for Systematic reviews and Meta-Analyses) [18]. We performed a search of papers with the MeSH terms previously mentioned. A total of 88 papers were obtained, and 16 were excluded because they were letters to the editor, review articles or the lung was not the tissue of the microbiome study (Figure 2). In the bibliometric analysis with the keywords used in articles, we found that the main topics in lung microbiome studies were immunology response, host–pathogen interaction, dysbiosis, asthma, chronic obstructive pulmonary disease (COPD), cystic fibrosis (CF) and the gut–lung axis. The conceptual structure analysis using MDS (Figure 3) showed that within the lung microbiome research, two groups converge on the word “lung.” The first group (blue) contains terms related to infectious diseases (tuberculosis and pneumonia) and innate, adaptive, and respiratory mucosa immunity; the second group (pink) contains noninfectious diseases such as cystic fibrosis, lung neoplasms, obstructive diseases, and asthma.

### 3.1. Lung Microbiome

The health condition of the lung microbiome is a dynamic balance between oropharyngeal immigration (due to microaspirations, air inhalation, and direct dispersion along mucosal surfaces) and mucociliary clearance, host response mechanisms such as coughing and, to a lesser extent, the differential reproduction rate of microorganisms [19]. The lung microbiome in healthy subjects is characterized by a low bacterial load and a broad diversity of species; the most abundant genera are *Prevotella, Streptococcus, Veillonella, Neisseria, Haemophilus,* and *Fusobacterium* [4,6]. Changes in the structure of the healthy lung microbiome (called dysbiosis) influence an overactive immune response [20], the release of proinflammatory cytokines, alterations in translational pathways, recruitment, cellular activation [21], accelerated tissue aging [22] and autoimmunity phenomena [9,21].

The lung microbiome composition is also dysregulated by tobacco smoking [23]. It has been observed that there is a positive correlation between the years of cigarette smoking and taxa of the phylum *Firmicutes* (genera *Veillonella* and *Megasphaera*) and the genus *Prevotella*, while this variable is inversely associated with the taxon *Proteobacteria* (genera *Eikenella* and *Haemophilus*) [24]. Therefore, it is essential to consider the composition of the pulmonary microbiome as one more factor that contributes to the development of diseases and even the severity with which they occur.

### 3.2. Lung Microbiome and Immune Response

There is no clear division between the innate and adaptive response since they are not isolated pathways that can act without the other (Figure 4).

The innate immune response is the first body defense barrier mechanism against potential pathogen agents; this pathway is activated by the presence of particulate matter, toxins, allergens, microorganisms, and endogen detritus as dead cells that come from the environmental air [25]. The lung innate immune response is integrated by mechanical elements, and the respiratory epithelium is one of the most important components. It is pseudostratified and formed by goblet cells with ciliary modifications. Additionally, found in the nonspecific immune response are dendritic cells, alveolar macrophages, neutrophils, and NK-cells; chemokines and interleukins such as CXCL-1–8, CXCL12, CCL2, CCL17, CCL18, CCL20, IL-1α, IL-1β, IL-10, IL-17, IL-23, IL-25, IL-33 and thymic stromal lymphopoietin; Toll-like receptors (TLR), RIG-I-like receptors (RLP), Nod-like receptors (NLR) and C-type lectin receptors (CLR) [26,27]; and even soluble molecules (lysozyme, defensin, and complement proteins) [28]. The epithelium has been associated with multiple mechanisms of interaction with the pulmonary microbiome, has a poorly permeable barrier function and can sense microorganisms and respond to their presence [29]. The bacterial clearance function is given by the brushing function of microvilli and cilia; when their mobility is altered, an increase in the bacterial load is observed, producing infectious diseases, as occurs in mucociliary dyskinesia [30]. This mechanism is facilitated by the entrapment of bacterial agents in a mucous solution rich in antibacterial products, whose changes in density and composition lead to the inappropriate growth of bacteria [27]. Furthermore, if the integrity of the pulmonary epithelium is interrupted by pathogens, detected by molecular recognition mechanisms, a series of anti-inflammatory and inflammatory mechanisms are activated, which will be further explored later [31].

The adaptive immune response includes the participation of specialized cells (cell immunity) and immunoglobulins (humoral immunity); this response is dynamic and depends on the exposure of the organism to exogenous agents, as well as the microbiome composition and metabolites and the local microenvironment [10].

Dendritic cells are pivotal components of the processing and presentation of antigens to adaptive immune cells. The adaptive immune response is classified into three types [10]. The adaptive immune response classified as type 1 or T-helper type 1 (Th1) is characterized by the production of IFN-γ, IL-2, TNF-α, and GM-CSF; this pathway activates macrophages that help in intracellular pathogen clearance and stimulates the production of IgM, IgG, and IgA from B cells. Adaptive immune response type 2 (Th2) is characterized by the production of IL-4, IL-5, IL-9, IL-10, and IL-13, which promote IgE and IgG antibody production. Type 3 immunity integrates RORγt^+^ lymphocytes that produce IL-17A, IL7F, IL-22, and IL-26, and this pathway is activated in the presence of bacteria or extracellular fungi [32,33].

The lung microbiome influences the host immune response at the local level in murine models [25]; when the commensal microbiome is absent, antibacterial activity performed by alveolar macrophages through reactive oxygen species (ROS) production is compromised, endangering physiologic clearance of potentially pathogenic bacteria [34]. In mice receiving antibiotics and inoculated with *S. pneumoniae* and *K. pneumoniae*, the microbiome promotes a widespread innate response to airway infection by pathogens, stimulates clearance, and enhances host survival during infections, most likely due to a signaling axis that involves IL-17 and GM-CSF; last is a key factor for immunological regulation that prevents colonization by pathogens and has great importance in allergy mechanisms [35]. Furthermore, IL-17 is associated with the best bacterial clearance of species such as *S. pneumoniae* because of its functions in macrophage and neutrophil recruitment [36]. In healthy mice, IL-1α, a crucial cytokine for innate immunity in lung defense against bacteria, has a negative correlation with the presence of pathogenic bacteria and the diversity of bacterial communities [25]. In an in vitro lung fibroblast model of patients with idiopathic pulmonary fibrosis, higher species accumulation curve richness was significantly associated with the inhibition of nucleotide-binding oligomerization domain (NOD) and TLRs, whereas an increased abundance of *Streptococcus* correlated with increased NOD-like receptor signaling. [37]. Murine models have also demonstrated that commensal bacteria in the upper airways protect mice from death secondary to influenza virus infection by macrophage M2 polarization, which secretes anti-inflammatory mediators such as IL-10 and TGF-β [32]. The host immune response is not just mediated by microbiome colonization; even metabolites that are produced by the microbiota are implicated in the immune response; for example, human dendritic cells exposed to *P. aeruginosa* produce high levels of putrescine, which induces TNF-α, IL-6, and IL-10 [38].

For the commensal microbiome to coexist in an environment that is monitored by the immune system, the mucosal surface must be in an immunosuppressive state, and the communication between dendritic and epithelial cells is vital for maintaining balance. Retinoic acid and TGF-β produced by epithelial cells act in dendritic cells to make them tolerogenic to bacterial stimulation [39]; nevertheless, this balance can be affected by microbiome metabolites that participate in the mechanism of immune tolerance [40].

### 3.3. Lung Microbiome in the Most Studied Respiratory Pathologies

#### 3.3.1. Cystic Fibrosis (CF)

The respiratory microbiome in these patients was one of the first to be studied [41]. The pulmonary pathophysiology in these patients is characterized by chronic respiratory tract infection accompanied by an uncontrolled proinflammatory state. DNA-based analyses suggest that the diversity of lung microbiota may be a key disease parameter. Many studies have found that the relative abundance of nonconventional organisms and the diversity of lung microbiota in airway samples decrease with age and disease severity [42]. Through the analysis of the bacterioma in children with CF, pathogenic species such as *Prevotella denticola, Lysobacter sp, Tropheryma whipplei*, and *Granulicatella elegans* were identified [43]. Additionally, the decrease in microbial diversity is associated with a decrease in lung function [44]; even with greater severity of the disease, *Pseudomonas aeruginosa* predominates, with is a pathogen that produces proteases that contribute to tissue damage [45]. In adults with CF, the microbiome has been observed to be more stable, despite the use of antibiotics to *Streptococcus, Prevotella, Rothia*, and *Veillonella* found in this group of patients; in addition, the decrease in bacterial richness is associated with worse respiratory function [46]. The changes in the composition of the lung microbiome in patients with CF and the alterations that originate in the pulmonary ecosystem may be the cause of the low effectiveness of antibiotics in these patients [47]. Through metagenomic studies, it is possible to identify the dominant pathogens, as well as their resistance to antibiotics, quickly and accurately compared to bacterial cultures [48], contributing to improving the quality of life of patients with CF.

#### 3.3.2. Asthma

Asthma is a chronic and heterogeneous respiratory disease involving eosinophilic and noneosinophilic inflammatory pathways; in recent decades, despite new therapeutic options, improved sanitary conditions, and access to the health system, its prevalence has increased in industrialized countries; although the hereditary component has an important role, a change in the pattern of expression of the disease has been observed, suggesting a gene–environment interaction [49]. In 1989, Strachan introduced the “hygiene hypothesis,” a concept that has regained popularity in recent years, which proposes that the increase in atopic disease prevalence is influenced by the reduction in exposure to microorganisms that colonize the host in childhood and that this is responsible for a decrease in the microorganisms that compose the host microbiome, in addition to rising antibiotic intake. Various authors find this complex phenomenon to be an important factor in the increased incidence of asthma in occidental countries that share lifestyle characteristics, a diet deficient in fiber, sedentary lifestyle, and increased time indoors with less exposure to environmental allergens, viruses, and bacteria [50,51]. The hygiene hypothesis views the microbiome as a protective factor in the development of atopic diseases [52].

Longitudinal studies in humans confirmed that there is an inverse relationship between microbial diversity and the presence of allergens (acarine and house dust) that increases susceptibility to hypersensitivity phenomena such as asthma and rhinitis [49]. The microbiome in patients with asthma is different from that of healthy subjects, with an increase in the abundance of the genera *Haemophilus, Neisseria,* and *Moraxella* in comparison with healthy subjects [49,53]. The increase in the abundance of the genus *Proteobacteria* has been observed in severe forms of the disease [54,55].

The asthma microbiome composition depends on the airway (eosinophilic or neutrophilic) inflammation level. Colonization of the airway with *Moraxella catarrhalis* and *Haemophilus influenzae* is associated with increased levels of neutrophils and IL-8; however, a higher eosinophil count in the submucosal airway results in a lower relative abundance of *M. catarrhalis* [51]. The abundance of *Moraxella* and *Streptococcus* in BALF has a negative correlation with the abundance of *Corynebacterium*, and there is less expression of proinflammatory cytokines at the lung level (IL-6, IL-7, and IL-21). Disturbances in the composition of bronchial bacterial communities seem to contribute to the phenotype in which asthma occurs [56].

#### 3.3.3. Chronic Obstructive Pulmonary Disease (COPD)

COPD is a condition with progressive, nonreversible airflow limitation and pathological changes in the lung, among other mechanisms, due to alterations in the immune response; although the known pathological changes are mainly due to damage from tobacco smoking, biomass-burning smoke exposure and, less frequently, alpha-1 anti-trypsin deficiency, despite smoking cessation, the disease progression is not entirely understood. A genetic component is attributed that predisposes individuals to higher sensitivity to environmental agents; however, the microbiome–host interaction influences the progression of these pathological changes as well as the immune response [57]. Studies involving next-generation sequencing (NGS) have demonstrated that the lung microbiome in subjects with COPD promotes an inflammatory state dependent on the identified bacterial communities. The bacterial profile enriched in the *Prevotella* and *Veillonella* genera is associated with a Th17 pattern, with a higher presence of CD4 and IL-17 cells and the activation of TLR-4. At the same time, the phenotype composed of *Acidonella* and *Pseudomonas* has a positive correlation with the presence of macrophages and INF-γ [58].

Tissues of patients with COPD with an increased abundance of members of the *Bacteroidetes* and *Firmicutes* phyla present more significant alveolar destruction and increased IL-17 [59]. In the COPD murine model, there is an increased bacterial load and alterations in epithelial remodeling secondary to NF-κB factor activation, increased leukocyte infiltration, and increased expression of neutrophil elastase, causing emphysema-like changes in the lung parenchyma [56]. Microbial alterations also include changes in both the abundance of specific genera and respiratory function; the decrease in the *Treponema* genus and the increase in *Pseudomonas* have been associated with a reduction in respiratory function even in early stages of the disease [60].

The exacerbations, which have a significant impact on the decline of respiratory function, quality of life, hospitalizations, and intensive care unit attention, with a mortality rate close to 15%, are one of the main concerns in COPD [61]. Almost 70% of exacerbations are due to an infection, with bacteria being responsible for half of them. The main bacteria implicated are *Haemophilus influenzae*, *Streptococcus pneumoniae*, and *Moraxella catarrhalis* [62]. During the stable phase, longitudinal studies have identified a decrease in bacterial diversity and alteration in the systemic inflammatory response in pathways dependent on the bacterial response, such as TNF-α, IP-10, and MIG [63]. The use of antibiotics, such as azithromycin, usually used in COPD therapy to decrease the number of exacerbations, generates a decrease in bacterial diversity and an increase in microbial metabolites (glycolic acid, linoleic acid, and indole-3-acetate), resulting in altered expression of cytokines that regulate the immune response, such as IL-12, IL-13, CXCL-1, and TNF-α [40,45,64].

#### 3.3.4. Pulmonary Microbiome in Interstitial Lung Diseases

Interstitial lung diseases (ILDs) are a heterogeneous group of disorders that share radiologic and pathological features characterized by alterations in tissue repair mechanisms, resulting in fibrous tissue accumulation in the lung interstitium [65]. The ILD with the highest incidence is idiopathic pulmonary fibrosis (IPF) [66]. In murine models, *Bacteroidetes* and *Prevotella* were associated with higher secretion of bacterial outer membrane vesicles (OMVs), and these were related to an increase in IL-17 (A, B, and F) and TNF-α, promoting proinflammatory and profibrotic gene expression [67]. The genera *Staphylococcus, Prevotella*, and *Streptococcus* predominate in patients with IPF, which has an impact on the decrease in the expression of genes that participate in innate immune defense [37].

In the COMET-IPF cohort, patients were monitored over 80 weeks, and it was shown that the genera *Staphylococcus* and *Streptococcus* are related to the highest risk of disease progression [68]. Additionally, exacerbations have been associated with a decrease in *Veillonella* and an increase in *Stenotrophomonas* and *Campylobacter*. The latter genus is a gastrointestinal pathogen, proposed as translocation of the intestinal microbiome to the lung, which contributes to exacerbations in these patients [69]. This evidence has led to the hypothesis that increased lung bacterial load in IPF is secondary to microaspirations, and this association with gastroesophageal reflux also contributes to lung damage, coupled with the slow bacterial clearance of these patients because of mucociliary alteration [70]. Again, in the COMET-IPF study, it has been reported that the decrease in the diversity of bacterial communities in the lung is significantly associated with an increase in proinflammatory cytokines and growth factors (IL-1β, CXCL8, MIP-1α, G-CSF, VEGF and EGF) [71]. In an Asian population, it has been reported that the decrease in bacterial diversity is correlated with IPF progression, low forced vital capacity, and a reduction in the six-minute walk test. These data suggest that loss of microbial diversity was associated with disease activities of IPF [72].

Hypersensitivity pneumonitis is another ILD that very few studies have explored; in this disease, there is an increase in the *Staphylococcus* genus, and patients have lower bacterial alpha diversity compared to those with IPF [73]. In sarcoidosis, just a few years ago, *Atopobium* spp. and *Fusobacterium* spp. were identified in the highest abundance [72]. Unfortunately, there is no follow-up study of the microbiome in this disease.

It has been reported that 10% of patients with rheumatoid arthritis (RA) have lung injury compatible with ILD [74]. Patients in early RA with lung affliction have a lower abundance of the genera *Actinomyces*, *Burkhordelia* and *Prevotella*; in general, this dysbiosis is similar to that reported in sarcoidosis [21].

Another pathology of autoimmune origin associated with effects on the respiratory system is Wegener’s granulomatosis (WG), which is a systemic vasculitis with anti-neutrophil cytoplasmic antibodies (ANCAs), frequently affecting the upper respiratory system and having few effects on the lower airways. Evaluation of the nasal microbiome found that patients with WG have a low relative abundance of *Propionibacterium acnes* and *Staphylococcus epidermidis*; in patients with active disease, there is less abundance of the *Malasseziales fungus* compared to patients in remission [75].

#### 3.3.5. Pulmonary Microbiome in Lung Infections

Respiratory lung infections are a global health problem, and the host response to several pathogens is linked to diverse immunological mechanisms. In a murine model compared with and without dysbiosis due to inoculation with pathogens (*S. pneumoniae* and *K. pneumonia*), regulatory participation of the microbiome against bacterial infection was observed as an IL-17 release that activates GM-CSF and NOD-like, impacting this dysbiosis in rodent mortality [35]. In humans, it is suggested that the mechanism in which a bacterial infection occurs in the lower respiratory tract leading to pneumonia depends on alterations in the microbiome in the upper respiratory tract (URT) that allow dysbiosis with the increase in the abundance of pathogens to which the infection is attributed. It has been observed that pharyngeal colonization by *S. pneumoniae, H. influenzae,* or *M. catarrhalis* in healthy neonates is associated with an increased risk of presenting pneumonia and bronchiolitis during the first three years of life, demonstrating the participation of the microbiome in the maturation of the immune response against pathogenic bacteria [76]. In young and older adults, the dominance of three bacterial genera at the oropharyngeal level was associated with pneumonia, namely, a well-known causative agent *S. pneumoniae*, and two agents not described previously, *Lactobacilli* and *Rothia*, but their presence has a high sensitivity and specificity for pneumonia diagnosis [77]. A prospective study observed that in adult patients admitted with pneumonia to the intensive care unit (ICU) who required mechanical ventilation (MV), a decrease in bacterial diversity was associated with higher mortality, regardless of the bacterial composition, the reason for hospitalization or comorbidities [78]. In a prospective double-blind study in which the oral administration of intestinal probiotic preparations that contained *Lactobacillus* spp., *Bifidobacterium* spp. and *Streptococcus thermophilus* reduced the ICU stay time, the authors hypothesized that probiotic administration could prevent respiratory colonization of multidrug-resistant pathogens (MDR), a usual UCI-related infection that extends the need for mechanical ventilation and increases mortality. The mechanism that explains this protective effect of probiotics in the lung needs further studies to be defined, and to our knowledge, studies about the direct effect of the local microbiome remain unknown in this group of patients [79].

The microbiome also has a wide role in the lung response to viral infection; antibiotic (ampicillin, neomycin sulfate and metronidazole) induced intestinal dysbiosis in rodents has effects on pulmonary immunocompetence towards the response to infection with influenza A virus [80]. Studies in murine models then demonstrated that intranasal administration of *Lactobacillus* spp. can regulate the pulmonary inflammatory response to virus by inducing IgA, IL-12, cytotoxic T lymphocyte-stimulating cytokine, and NK cells, thereby promoting viral clearance [81]. In humans, a prospective study showed that early colonization by *Streptococcus, Moraxella,* and *Haemophilus* presents a higher risk of developing URT infection by viruses compared to those infected with *Cornebacteria, Staphylococcus*, and *Alloiococcus* [82]. In a multivariate analysis, the increased abundance of *Moraxella* in children was associated with both upper and lower respiratory tract infection by the main pathogenic agents of childhood pneumonia, *Rhinovirus*, *Enterovirus*, and Respiratory syncytial virus (RSV) [83]. This response of the microbiome as a defense against RSV infection has been determined to be associated with bacterial products derived from acetate, which stimulates the production of interferon type-1 by activating G-protein coupled receptor 43 (Gpr43) in epithelial cells, producing IFN-β and protecting the respiratory epithelium from viral infection by decreasing the bacterial load [84]. Germ-free rodents had a delayed immunological response secondary to a dysregulation in the activity of NF-kB that allows greater colonization of the opportunistic fungus *Cryptococcus gattii* [85]. In patients with HIV with bacterial pneumonia, it was observed that the predominance of a pattern with a greater abundance of *Prevotella* is associated with greater colonization of *Aspergillus* due to an increase in the expression of IL-17 in these subjects, resulting in a dysregulation of Th17 cells [86]. In a study of patients admitted to the ICU, it was observed that microbiome alterations do not influence the presence of *Candida* spp. However, the presence of these alterations has been associated with dysbiosis, which allows rapid colonization of this opportunistic pathogen [87].

Tuberculosis infection (Tb) is a highly contagious infection caused by mycobacteria of the *Mycobacterium tuberculosis* complex. Eradication of infection is still a World Health Organization priority due to the high resistance to anti-tuberculosis drugs and high morbimortality worldwide. In murine models with induced pulmonary dysbiosis, it has been observed that the decrease in a bacterial load before inoculation of the bacillus is associated with a higher bacillus load secondary to a decrease in IFN-γ and TNF-α, as well as an increase in Treg FoxP3+ lymphocytes. [88]. This study demonstrates that the pulmonary microbiome plays a role in resistance to infection by *M. tuberculosis*. Despite infection with the bacillus, the core microbiome of the patients did not differ with respect to its members compared to healthy subjects, but it did differ in relative abundance and prevalence. In this study, members of *Prevotella, Neisseria, Streptococcus, Lactobacillus* and *Parvimonas* were more abundant in patients with TB, while *Granulicatella* and *Aggregatibacter* were more abundant in healthy controls [89,90]. Further studies have also demonstrated that the lung microbiome in humans is associated with various states of Tb, with a greater abundance of *Pseudomonas* associated with treatment failure [80]. In patients with HIV infection, the presence of a bacterial pattern with the dominance of *Pseudomonas* was observed to be associated with increased detection of *Mycobacterium* spp. [86].

## 4. Discussion

The characterization of the respiratory microenvironment will allow us to understand how dysbiosis contributes to the development, prognosis, or presentation of worse patterns of disease; this knowledge will enable us to identify new therapeutic options to improve the management of pathologies. In gastrointestinal diseases, therapies related to restoring the microbiome composition balance have been used, ranging from antibiotic modification [91] to changes in its composition through molecules, such as carbohydrates, that promote the growth of specific bacterial communities (prebiotics), as well as supplements of live bacteria to maintain the ecological balance of the microbiome (probiotics) or by direct inoculation of biota from healthy individuals [92].

In the lung, these strategies are beginning to be explored. In a murine model, the administration of aerosolized *Lactobacillus rhamnosus* significantly reduced lung metastasis when compared to the group of mice that had been administered saline solution [93]; previously, it has been reported that intranasal inoculation of *L. rhamnosus* in mice decreases IL-6 levels at the pulmonary level and protects against infection by influenza viruses [81]. In humans, it has been shown that recolonization of the respiratory tract can be useful in several pathologies. In children with repetitive acute otitis media, the intranasal administration of *Streptococcus salivarius* 24SMB proved to be safe and effective in reducing the recurrence of otitis episodes compared to those children who were not inoculated with the agent [94]. However, the use of prebiotics or probiotics in lung diseases remains to be explored to determine if they are viable and safe for use in humans.

## 5. Conclusions

The lung, under physiological conditions, is inhabited by microorganisms (bacteria, viruses, and fungi) that are important in maintaining homeostasis, regulating immunological mechanisms, and participating in the development of certain diseases. The microbiome study made it possible to understand its participation in pathophysiological processes, as well as the participation in these processes by some of its metabolites. However, further studies are required to verify whether the microbial profile can be used as a biomarker of disease progression or poor prognosis and to identify new therapeutic targets that restore lung dysbiosis safely and effectively in lung diseases.

## Figures and Tables

**Figure 1 microorganisms-08-01059-f001:**
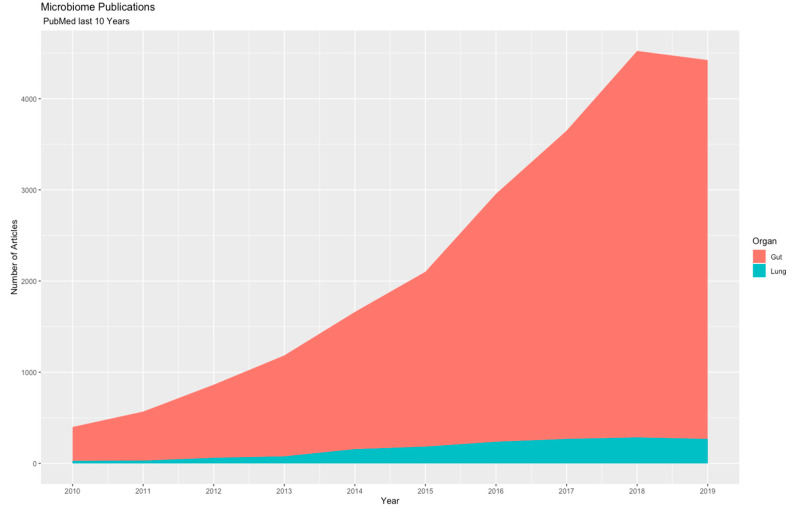
Frequency of accumulated publications in the gut and lung microbiome reported between 2010 and 2019.

**Figure 2 microorganisms-08-01059-f002:**
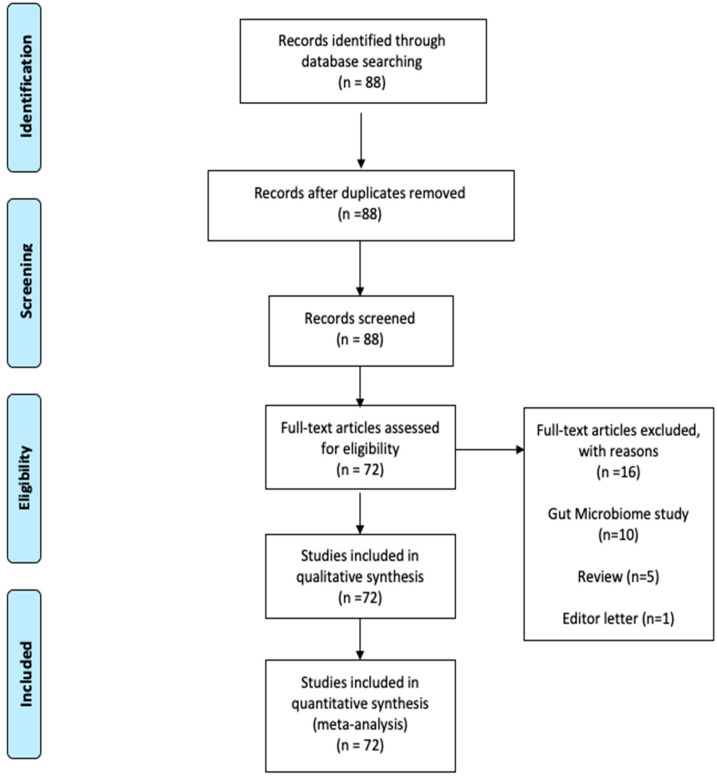
The article selection process was carried out following the PRISMA guidelines (Preferred Reporting Items for Systematic reviews and Meta-Analyses) [18].

**Figure 3 microorganisms-08-01059-f003:**
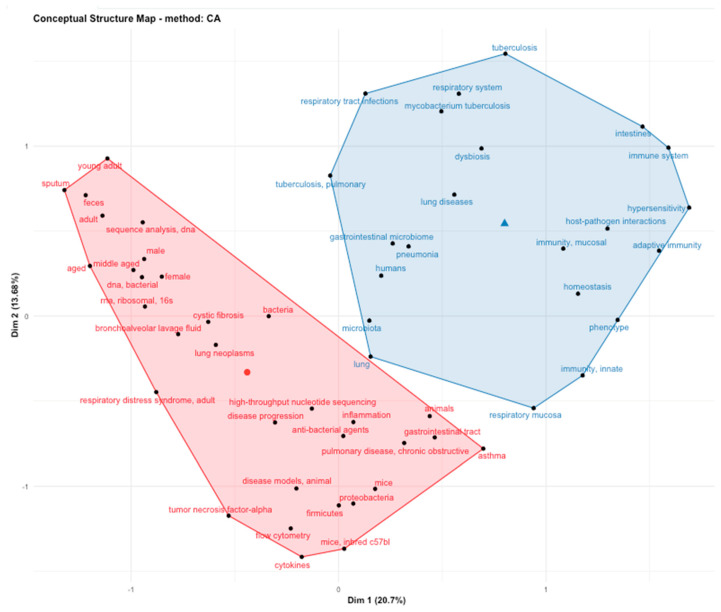
Analysis of the conceptual structure by multidimensional scaling (MDS) of the keywords included in the articles analyzed in this review.

**Figure 4 microorganisms-08-01059-f004:**
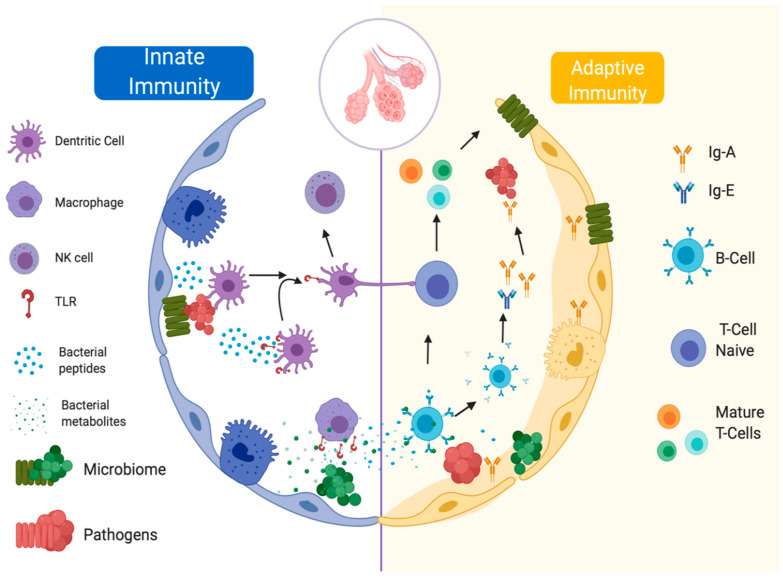
Main actors of the innate and adaptive immune response and the lung microbiome.

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
