# Peer review of "Lung Microbiome Participation in Local Immune Response Regulation in Respiratory Diseases"

_microorganisms, 2020, doi:10.3390/microorganisms8071059_

Round 1
Reviewer 1 Report
The review covers a novel and relevant topic on the role of the lung microbiome in health and disease, especially regarding the interaction with the immune system. I propose some modifications to improve the manuscript and some minor comments.
Major comments
English should be checked because some sentences were too long and difficult to follow. Try to split some of the long ones in two.
In Material and Methods, the word "microbiome" should be included since microbiota and microbiome are used indistinctly in this kind of study. It would improve the search for papers.
I would add a section about the role of the lung microbiome in infectious diseases such as pneumonia. The interaction between the immune system and the microbiome seems crucial for the outcome of the disease.
Lines 156-161: This aspect of the equilibrium between the microbiome, the epithelium, and the immune system is exciting. It would improve the review to develop a little more this idea.
Minor comments
Introduction
Lines 35-36 Include archaea, and other eukaryotes besides fungi (such as protozoa) since their role in physiology as part of the microbiome has started to be considered.
Line 37: Change "his" to "its."
Line 49: Change "micome" to "mycobiome."
Line 49: The sentence sounds strange. Change it to "It also participates in host protection against pathogenic microorganisms."
Line 51: Change "have" to "has."
Line 52: Change "won't" since it is not very formal.
Line 53: Change "like been" to "like being a closed."
Line 59 Change "is short in comparator on of other ecological niches like gut" to "is short compared to that of other ecological niches like the gut."
Line 61: It is "PubMed"
Line 62: Change "are" to "were."
Line 130: Change to the plural "macrophages."
Lines 126-131: The way to express this information is unclear, please re-write the sentence.
Line 143: Change to "associated with"
.
Lines 146-150: Re-write since the idea of the sentence is not clear. Is it because the "Prevotella" and "Streptococcus" are members of the microbiome? Clarify the role of the microbiome in this context.
Lines 163-180: As far as I know, cystic fibrosis also in BAL has been possible to detect pathogenic microorganisms. Check studies: i.e.
https://www.sciencedirect.com/science/article/pii/S2211124719304255
Line 168: Change "bacterioma" to "bacteriome."
Line 182-186: The sentence is too long and difficult to follow, re-write that sentence.
Lines 186-191: Change to "Since 1989, Strachan has introduced "Hygiene hypothesis," which has been retaken in recent years. This concept proposes that the increase of atopic disease prevalence is influenced by the reduction in the exposition to microorganisms that colonizes host in the childhood, and this is responsible for a decrease in the microorganisms that composes host microbiome, in addition to rising antibiotics intake."
Line 222: Change "While" to "At the same time."
Line 224: Add "the" to tissue.
Lines 263-265: The result from the Asian population study is not easy to follow, please re-write. What is the meaning of "6-minute walk distance". Please, clarify.
Line 271: Change "has" to "have."
Lines 280-284: Change "will allow understanding" to "will allow us to understand."
Line 284: Remove an extra "." from the sentence.
Line 293: Change "microbiome" to "tract." It is the place that is recolonized.
Line 297: Change "that they are" to "if they are."
Lines 299-301: Remove the second and third "in"; they are not necessary.
Lines 302-303: Write the sentence again. Metabolite's participation sounds strange.
Author Response
Thank you for comments. We have tried to address all the point. We add pdf file

Reviewer 2 Report
Major comments:
The given review performs meta-analysis of lung microbiome related work and summarizes the current understanding of role of lung microbiome in lung disease. The work is enlightening but needs significant revisions as suggested below. Firstly, the work describes role of microbiome in multiple lung diseases but the authors have explicitly stated "interstitial lung disease" in the title. The title will need rewording to more accurately reflect content of the review. In multiple places, the authors have used long, difficult-to-read, comma-filled sentences that lack necessary citations. sentences will need to be broken apart and appropriate citations will needed to be provided. Certain figures will also need major revisions since they do not exactly portray accurate methods or biology. Finally, extensive editing of the english language (beyond ones suggested below) will be required.
Specific comments:
Line 18 “adaptive” (Also occurs in multiple places in the manuscript. Please make necessary changes)
Line 37 “analyze its”
Line 40 …. “nevertheless, since 2003, there was evidence that the respiratory tract of patients with cystic fibrosis hosted bacteria”
Line 41 “Afterwards..”
Line 44 “… from patients with lung disease” (asthma may or may not be due to lung infections and all the patients in the cited study #4 were free from clinical infection)
Line 44-47 This sentence is not understandable. Are the authors saying that “lung is an organ that is exposed continuously to microorganisms that enter by inhalation process and that oral taxa similarity with healthy subjects can be measured by comparing airway tract microbiome sampled from spittle aspirations”?
Line 47 supine position
Line 48 “microorganisms like viruses (virome) [8], fungus (mycome), and bacteria (bacteriome) have a role in homeostasis [9] like participation…..”
Please note it is mycome and not micome.
Line 51 “immunological barriers”
Line 53 “….shares feature with these ecologic niches regardless of their particular characteristics like being an enclosed mucous tissue system,…..”
Line 59 “….in comparison to other ecological…”
Line 60 “…gut microbiome and associated disease were available in PubMed….”
Line 63 “Considering the importance..”
Line 80 “Meta-Analyses”
Line 80 “A search of papers with the MeSH terms previously mentioned provided a total of 88 papers were of which 16 were excluded because they were letters to the editor, review articles or lung was not the tissue of microbiome study (figure 2)”
Figure 2 needs more clarification. For eg how there can are n=72 records eligible for this study despite exclusion of n=88 records from total of n=88 records screened is unexplained. I assume the authors meant to move the box with n=16 excluded articles beside the n=88 screened records which will leave n=72 eligible records.
Line 86 “.. showed that within lung microbiome research…..”
Line 98 “host response mechanisms”
Line 101 “Haemophilus and Fusobacterium”
Line 101 “..structure of healthy lung”
Line 102 “…influences an…”
Line 105 Not sure what the authors are trying to suggest. They start by suggesting that tobacco smoking and other environmental factors cause dysregulation on lung microbiome and then suggest use of microbiome as a factor contributing to disease. It would be helpful if authors cite more solid articles showing these causative links. The studies cited for tobacco, pollution and age suggest the effect of environmental factors on microbiome and do not suggest that microbiome contributes to disease severity.
Line 105 the sentence needs rewording to explain the authors of line of thought more clearly and smoothly.
Line 114 “particulate”
Line 123 “…depends on exposure of the organism to exogenous agents such as….”
Line 129 “activates”
Line 130 “..stimulates production of IgM, IgG and IgA from B cells”
Figure 4 the figure seems to suggest that DCs are presenting antigens to B cells? If my reading of the figure is wrong and the authors suggested another biological interaction of DCs and B cells, adding a figure legend describing what the schematic represents is necessary.
Line 132 The response the authors are describing is classified as Th17 response and not a type 3 response.
Line 138-141 Citation/s to support this statement is missing
Line 152 “anti-inflammatory mediators”
Line 158 “retinoic acid”
Line 169 use italics to mention species names (please make similar changes throughout the document)
Line 170-175 These sentences are too long and include use of commas when a period can suffice and get the message across more easily. Please break the sentences to smaller sentences.
Line 182 “Asthma is a chronic….”
Line 182-186 Again a very long, comma-filled sentence. Please break into smaller pieces.
Line 182-186 please include appropriate citation
Line 186 “In 1989, Strachan introduced “Hygiene hypothesis”, a concept that has regained popularity in recent years and proposes that….”
Line 189 “exposure” (please make similar changes throughout the document)
Line 204-205 needs rewording
Line 211-217 Again a very long, comma-filled sentence. Please break into smaller pieces.
Line 224 “Tissues of COPD patients….”
Line 226 “bacterial load”
Line 234 “…bacteria being responsible for half of them…”
Line 249 “promoting”
Line 266 “… is another ILD very few studies have explored”
Line 273 “…associated with effects on the respiratory…”
Author Response
Thank you for comments. We have tried to address all the points. We add pdf file

Round 2
Reviewer 1 Report
The authors have addressed all the suggested changes. It is now excellent for publication.
Reviewer 2 Report
All my comments have been addressed.